# Enhanced Energy Absorption of Additive-Manufactured Ti-6Al-4V Parts via Hybrid Lattice Structures

**DOI:** 10.3390/mi14111982

**Published:** 2023-10-26

**Authors:** Seong Je Park, Jun Hak Lee, Jeongho Yang, Seung Ki Moon, Yong Son, Jiyong Park

**Affiliations:** 1Singapore Centre for 3D Printing, School of Mechanical and Aerospace Engineering, Nanyang Technological University, 50 Nanyang Avenue, Singapore 639798, Singapore; seongje.park@ntu.edu.sg (S.J.P.);; 2Additive Manufacturing Innovation Agency, Korea Institute of Industrial Technology, 113-58 Seohaean-ro, Siheung 15014, Republic of Korea; 3School of Mechanical Engineering, Pusan National University, 2 Busandaehak-ro, 63 beon-gil, Geumjeong-gu, Busan 46241, Republic of Korea; 4Advanced Joining & Additive Manufacturing R&D Department, Korea Institute of Industrial Technology, 156 Gaetbeol-ro, Yeonsu-gu, Incheon 21999, Republic of Korea; 5Department of Convergence Manufacturing System Engineering, University of Science and Technology (UST), 217 Gajeong-ro, Yuseong-gu, Daejeon 34113, Republic of Korea

**Keywords:** hybrid lattice structures, energy absorption, additive manufacturing (AM), powder bed fusion (PBF)

## Abstract

In this study, we present the energy absorption capabilities achieved through the application of hybrid lattice structures, emphasizing their potential across various industrial sectors. Utilizing Ti-6Al-4V and powder bed fusion (PBF) techniques, we fabricated distinct octet truss, diamond, and diagonal lattice structures, tailoring each to specific densities such as 10, 30, and 50%. Furthermore, through the innovative layering of diverse lattice types, we introduced hybrid lattice structures that effectively overcome the inherent energy absorption limitations of single-lattice structures. As a result, we conducted a comprehensive comparison between single-lattice structures and hybrid lattice structures of equal density, unequivocally showcasing the latter’s superior energy absorption performance in terms of compression. The single-lattice structure, OT, showed an energy absorption of 42.6 J/m^3^, while the reinforced hybrid lattice structure, OT-DM, represented an energy absorption of 77.8 J/m^3^. These findings demonstrate the significant potential of hybrid lattice structures, particularly in energy-intensive domains such as shock absorption structures. By adeptly integrating various lattice architectures and leveraging their collective energy dissipation properties, hybrid lattice structures offer a promising avenue for addressing energy absorption challenges across diverse industrial applications.

## 1. Introduction

In the automobile, ship, and aerospace industries, materials are required to be lightweight and have high mechanical properties because of fuel efficiency and the safety of passengers [1,2,3,4]. To address these requirements of high mechanical properties as well as lightweight, lattice structures have been used. Lattice structures have various characteristics due to their porous geometry compared to solid structures in terms of specific strength and energy absorption [5,6,7]. According to Maxwell’s stability criterion, lattice structures were classified into two types: stretch- and bending-dominant structures. Generally, the stretching-dominated structures represent higher strength and stiffness than bending-dominated structures. Bending-dominated structures are statically unstable and with high compliance [8,9,10].

In several studies, the effects of single-lattice structures on PBF parts were investigated with respect to mechanical strength and lightweight [11,12,13]. Based on characteristics that resist certain forces of lattice structures, a model with high strength and lightweight was devised [14]. Additionally, a shift block support for the turbine blade of a hovercraft was additive-manufactured (AMed) using lattice structures to reduce its weight in equivalent mechanical performance compared to solid parts [4]. In addition, lightweight injection molding was designed using lattice structures. Researchers fabricated the lightweight injection mold with a reduced weight of up to 79% using lattice structures, both stretch- and bending-dominant, and succeeded even in the injection molding test [5]. Automobile components also were optimally designed using lattice structures for AM while maintaining their original mechanical properties. Lightening the weight of automobile components it leads to positive effects in terms of fuel efficiency improvement [15,16]. In aerospace, in 2015, lattice structures were used to reduce the weight of satellites. The weight of the commutations satellite’s bracket was lowered by up to 35% compared to the previous bracket through lattice structures [17]. Furthermore, in the case of hybrid lattices with a combination of single-lattice structures, hybrid lattices were fabricated by inserting the unit cell of the lattice into the main single-lattice structure [18]. Another hybrid lattice structure was composed of a combination of two different single-lattice structures together in one unit cell [19,20]. There is also a study that implemented hybrid lattice structures by arranging the same single-lattice structure regularly with different densities [21]. When single-lattice structures with two different characteristics were mixed, the mechanical properties of all the hybrid lattice structures were improved.

Despite the increasing interest and application of lattice structures, several challenges remain. Previous research has often focused on the properties and benefits of individual lattice structures, but comprehensive investigations into their limitations, especially in real-world applications, are limited. There has been an increasing demand for structures that can combine the benefits of multiple lattice types, especially in high-impact environments where energy absorption is crucial. Moreover, while many studies have emphasized the advantages of single-lattice structures, there is a gap in understanding how combining different structures can potentially overcome inherent limitations or provide synergistic benefits. Thus, the fundamental studies that have structurally increased mechanical properties using a stacked lattice-by-lattice structure without collapse at the interface between different lattices are not enough.

Here, we structurally enhanced energy absorption using hybrid lattice structures for application in various industries. Under optimum laser conditions, the octet-truss, diamond, and diagonal lattice structures were AMed as single-lattice structures according to density. Furthermore, hybrid lattice structures were designed to break through the inherent limitations in terms of the energy absorption of single-lattice structures by stacking the lattice by lattice. Our study clearly demonstrated that hybrid lattice structures were superior to the single-lattice ones in terms of energy absorption despite having the same density. Thus, hybrid lattice structures can be used in crashworthy structures that require a lot of energy absorption.

## 2. Experimental Section

### 2.1. PBF-Based 3D-Printing Processes

The lattice structures were fabricated using a commercially available PBF system (DMP flex 350, 3D Systems, Rock Hill, SC, USA) and Ti-6A-4V (LaserForm Ti Gr23, 3D Systems, USA) powder, which had a mean particle size of 33 μm. All processes were carried out in an argon atmosphere within an oxygen concentration of 3 ppm to prevent the oxidation of Ti-6Al-4V. The laser power, scan speed, layer thickness, and hatching distance were determined, according to our previous research’s parameters, to be 125 W, 2800 mm/s, 30 μm, and 110 μm, respectively [5]. The scan strategy was set as a 10 × 10 mm^2^ chessboard pattern. The interlayer rotation angle was 67°.

### 2.2. Design for Lattice Structures

The lattice structures used were octet-truss (OT), diamond (DM), and diagonal (DG), as shown in Figure 1a. OT was chosen as a stretch-dominant structure. The OT has more struts in its unit cell than any other lattice structure. DM was chosen as a bending-dominant structure. Since the DM has a small number of struts, the thickness of the struts increased significantly when increasing the density, compared to other lattice structures. DG corresponds to bending-dominant structures theoretically according to Maxwell’s stability criterion. DG was chosen because it has z-axis struts that resist compression. All single-lattice structures consisted of OT, DM, and DG according to densities of 10, 30, and 50%, as shown Figure 1b. In particular, the internal space began to become clogged when the density reached 50% in the case of OT, so 50% was set as the maximum density [5]. The hybrid lattice structures were designed by mixing the OT, DM, and DG with a density of 30%, as shown in Figure 1c. Because OT is structurally stable with many struts in a unit cell, OT was placed at the bottom in the hybrid lattice structures for stable deposition. As shown in Figure 1a, OT has 5 nodes, such as the vertex and center in the upper surface. It is shown that these nodes can support the center node of DM and the vertex nodes of DG. All lattice structures were designed using 3DXpert 16 (3D Systems, USA) software and consisted of 1 × 1 × 1 mm^3^ unit cells in the compression specimens.

### 2.3. Compression Tests

The compression specimens were fabricated according to ISO 13314: Mechanical testing of metals—Ductility testing—Compression test for porous and cellular metals. The compression test was conducted at a cross-head speed of 2 mm/min using a universal testing machine (AGS-X 300kN, Shimadzu, Kyoto, Japan). All of the compression tests were performed using three specimens without post-processing. The compression strength was calculated by dividing the cross-section (12.7 × 12.7 mm^2^) of the specimen by the force displayed by the load cell of the universal testing machine. The value of energy absorption under the stress–strain curve becomes Equation (1):(1)$$W=\;\int_{0}^{\varepsilon_{m}}\sigma d\varepsilon$$
where ε, $\varepsilon_{m}$, and σ are strain, up to strain, and stress, respectively [5].

## 3. Results and Discussions

### 3.1. Compression Behavior of Single-Lattice Structures

Figure 2 shows the stress–strain curve for the compression test of the single-lattice structures with various densities (10, 30, and 50%). The yellow dots in the graph indicate the moment immediately after the start of the compression test, after plastic deformation, and after the end of the compression test, respectively. The specimen at the moment of the yellow dot is shown below the graph. The compression behavior of the lattice structures can be expressed as one of three: (i) the onset of plasticity, (ii) plateau stress, and (iii) densification [11,22].

Figure 2 shows the stress–strain curves of OT according to density. The stress–strain curves of OT for the 10, 30 and 50% densities represent the stress drop after the onset of plasticity points of approximately 45, 237, and 331 MPa, respectively. In general, the OT, stretch-dominant structures, show oscillation during plastic deformation in compression load [23,24]. In this study, the stress shows an oscillation curve at plateau stress because it corresponds to the partial fracture of the struts. The densification region is not noticeable in 10 or 30% density OT, while it is significant in the 50% density OT.

The stress–strain curves of the 10, 30, and 50% density DM show points of onset of plasticity at approximately 64, 138, and 257 MPa, respectively, as shown in Figure 3. The DM, bending-dominant structures, have significant plateau stress keeping the onset of plasticity-level stress without stress drop like OT [25,26]. The 10% density DM is brittle fracture after plateau stress, the 30 and 50% DMs show increasing stress after plateau stress as densification.

In the stress–strain curves of DG, as shown in Figure 4, the graphs of the 10 and 30% density DM show the onset of plasticity as the highest stress at approximately 133 and 244 MPa, respectively. In addition, the onset of plasticity of 10 and 30% appear blunt like DM, not sharp like OT. Furthermore, the plateau stress regions show short and insignificant behavior because the z-direction struts of the DG lead to early brittle failure during the compression test.

All lattice structures with a 50% density showed the phenomenon of plateau stress, and densification appeared simultaneously. Furthermore, they represent a strain shorter than the 30% density samples, except DG. In other words, because the density was relatively high, the structures were packed under the compression load, which leads to early brittle fracture. DG with a 50% density is ambiguous at the starting point of the onset of plasticity and has a behavior similar to the compression behavior of the solid structure. The value for ultimate compression strength (UCS) and energy absorption of all lattices is shown in Figure 5.

### 3.2. Compression Behavior of Hybrid Lattice Structures

As mentioned in Section 3.1, OT with a 10% density easily collapsed with a low UCS of 45 MPa and a low strain of 0.07. In addition, all lattice structures with a 50% density showed stress–strain behavior similar to that of a solid structure after the onset of plasticity. Lattice structures with a 30% density represented stress–strain curve behavior similar to that of lattice structures in terms of the onset of plasticity and plateau stress. In addition, lattice structures with a density of 30% have a low standard deviation in terms of UCS and the energy absorption between them. In the case of 30%, UCS and energy absorption showed standard deviations of 37.9 MPa and 3.9 J/mm^3^, respectively (standard deviations for UCS/energy absorption of 10% and 50%: 46.1 MPa/3.2 J/mm^3^ and 53.3 MPa/19.4 J/mm^3^). Accordingly, the density of the hybrid lattice structures was selected for the single-lattice structures with a 30% density to show the compressive behavior of lattice structures. Figure 3 represents an AMed hybrid structure and the stress–strain curves for the compression test of the hybrid lattice structures. In the 30% density single-lattice structure, the DG had the best UCS, while the DM had the weakest UCS. Accordingly, from the photograph of the specimen in the stress–strain curve, DM collapsed before OT, as shown in Figure 6a, and OT collapsed before DG, as shown in Figure 6b. The compression behavior of the hybrid lattice structures showed the single-lattice structures’ compression behavior in the order of the fracture of the single lattices composing the hybrid lattice structures. Thus, since the compression behavior of the two single lattices occurred sequentially, it showed a superior energy absorption [18].

### 3.3. Compression Behavior of Hybrid Lattice Structure with a Reinforcement

The stress–strain curves of the hybrid lattice structure with the added reinforcement exhibited behavior similar to that of the previous configurations. In the OT-DM hybrid lattice structure, preferential yielding of DM occurred before OT, as observed in the previous tests. Similarly, in the OT-DG hybrid lattice structure, OT yielded before DG, as shown in Figure 7. However, the key aspect of interest in this section is the energy absorption of the different configurations. The energy absorption values of the single-lattice structures, hybrid lattice structures, and hybrid lattice structures with added reinforcement were compared.

The results showed that the hybrid lattice structures with reinforcement displayed the highest energy absorption among the tested configurations, as shown in Figure 8. The solid center-plate reinforcement contributed to an additional increase in energy absorption, thereby further enhancing the mechanical properties of the hybrid lattice structures. These findings highlight the potential of controlling and manipulating the energy absorption properties of lattice structures through the addition of reinforcements. By modifying the geometry and characteristics of the reinforcement, it becomes possible to tailor the mechanical properties and energy absorption capacity of hybrid lattice structures to specific application requirements.

Overall, Section 3.3 emphasizes the significance of the reinforcement approach in enhancing the energy absorption capabilities of hybrid lattice structures. The results suggest that this approach can be effectively utilized to achieve desired mechanical properties and optimize energy absorption in lightweight structures for various industrial applications.

## 4. Conclusions

In this study, the authors investigated the compression behavior and energy absorption of single-lattice structures and hybrid lattice structures fabricated using AM with Ti-6Al-4V. The lattice structures were designed based on OT, DM, and DG. The main findings and conclusions of the study are as follows:Hybrid lattice structures, which combined different lattice types (OT, DM, and DG), demonstrated higher energy absorption than single-lattice structures of the same density. This indicates that the hybrid configuration overcomes the inherent limitations of single-lattice structures in terms of energy absorption.Single-lattice structures exhibited distinct compression behaviors depending on their dominant structural characteristics. OT structures, representing stretch-dominant structures, showed oscillation during plastic deformation. DM structures, representing bending-dominant structures, exhibited plateau stress without a stress drop, while DG structures showed the blunt onset of plasticity and early brittle failure due to z-direction struts.Among single-lattice structures, DG with a 50% density displayed behavior similar to that of solid structures after the onset of plasticity, indicating early brittle fracture. On the other hand, OT with a 30% density demonstrated the best UCS and energy absorption, while DM with a 30% density exhibited the weakest UCS among the 30% density structures.Hybrid lattice structures composed of OT-DM and OT-DG combinations exhibited the sequential collapse of the single lattices, leading to superior energy absorption compared to single-lattice structures. This indicates that hybrid configurations can enhance energy absorption capability by utilizing the specific properties of individual lattice types.To further enhance energy absorption, a solid center plate was added as reinforcement between the lattice structures. The enhanced hybrid lattice structures demonstrated the highest energy absorption among the tested configurations, highlighting the potential to control energy absorption by modifying the geometry of the reinforcement.

Based on these findings, it can be concluded that hybrid lattice structures fabricated using AM techniques have the potential to serve as lightweight and energy-absorbing structures in various industries, including the automotive, aerospace, and shipbuilding industries. The ability to control and optimize energy absorption through hybrid configurations and reinforcement opens up new possibilities for crashworthy structures that require high energy absorption capabilities.

## Figures and Tables

**Figure 1 micromachines-14-01982-f001:**
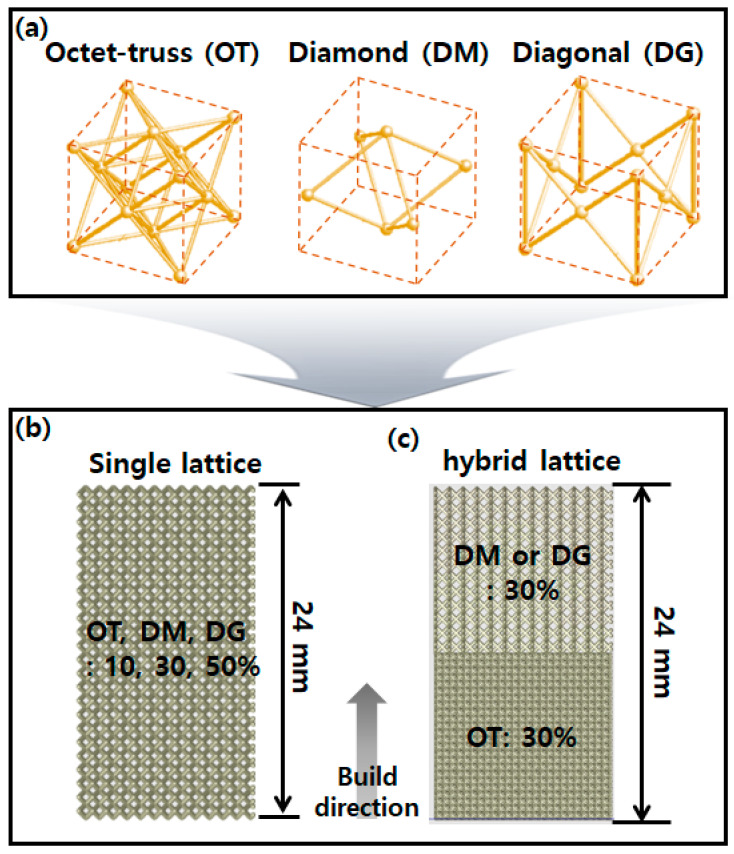
(**a**) OT, DM, and DG lattice unit cells. Dimension for compression specimen of (**b**) single-lattice and (**c**) hybrid lattice structures stacking them lattice by lattice.

**Figure 2 micromachines-14-01982-f002:**
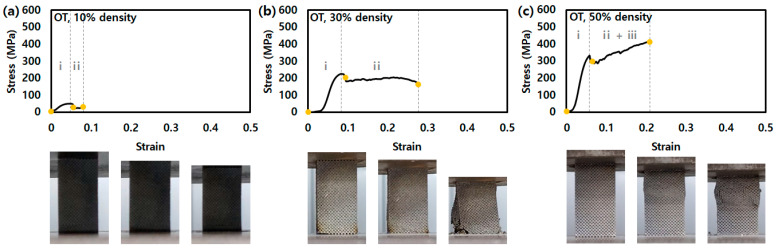
Stress–strain curves of lattice structures for OT with a density of (**a**) 10%, (**b**) 30%, and (**c**) 50% in compression test. (i: region before onset of plasticity point, ii: region of plateau stress, and iii: densification).

**Figure 3 micromachines-14-01982-f003:**
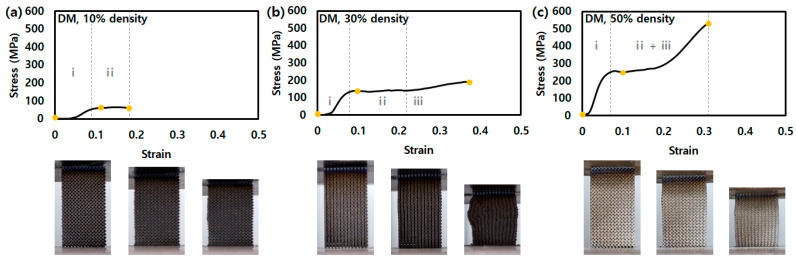
Stress–strain curves of lattice structures for DM with a density of (**a**) 10%, (**b**) 30%, and (**c**) 50% in compression test. (i: region before onset of plasticity point, ii: region of plateau stress, and iii: densification).

**Figure 4 micromachines-14-01982-f004:**
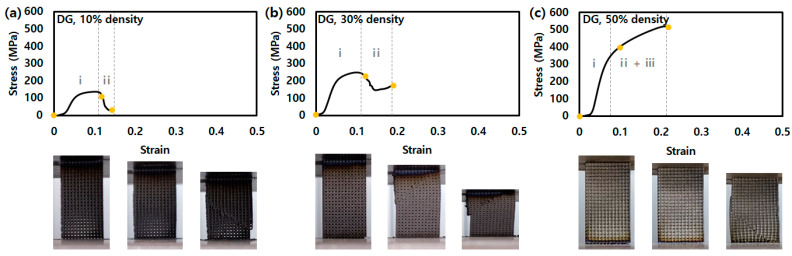
Stress–strain curves of lattice structures for DG with a density of (**a**) 10%, (**b**) 30%, and (**c**) 50% in compression test. (i: region before onset of plasticity point, ii: region of plateau stress, and iii: densification).

**Figure 5 micromachines-14-01982-f005:**
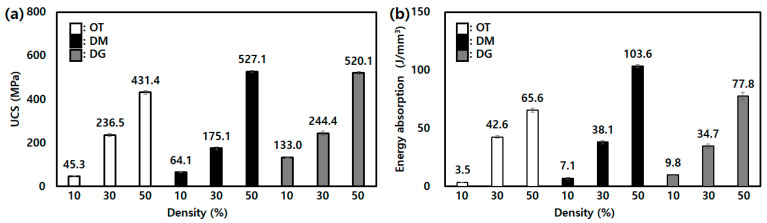
Comparison of results for (**a**) UCS and (**b**) energy absorption of single-lattice structures.

**Figure 6 micromachines-14-01982-f006:**
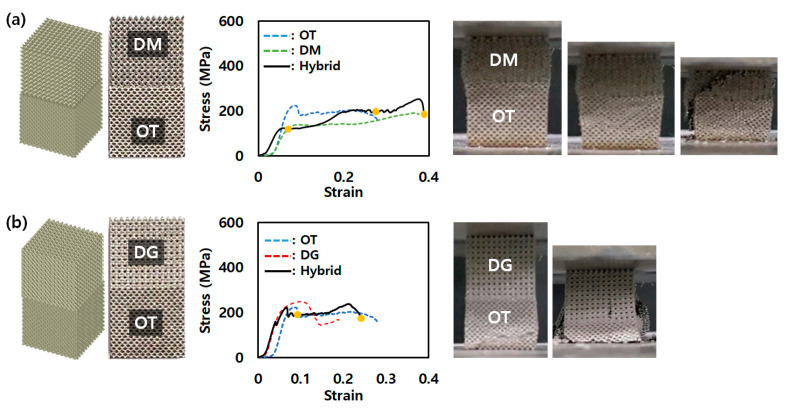
AMed hybrid lattice structures and stress–strain curves for compression test of (**a**) OT-DM and (**b**) OT-DG.

**Figure 7 micromachines-14-01982-f007:**
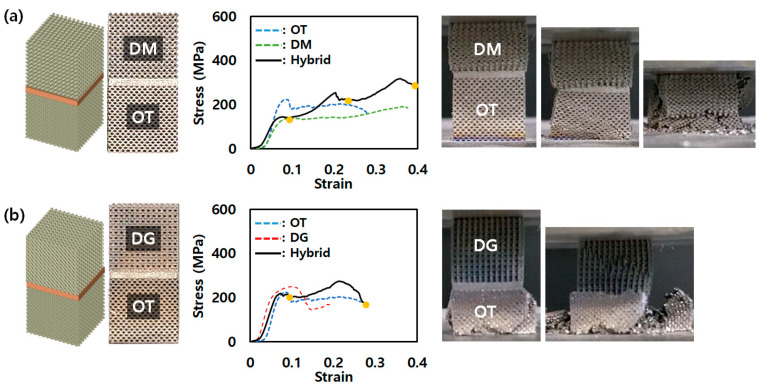
AMed hybrid lattice structures and stress–strain curves for compression test of enhanced (**a**) OT-DM and (**b**) OT-DG.

**Figure 8 micromachines-14-01982-f008:**
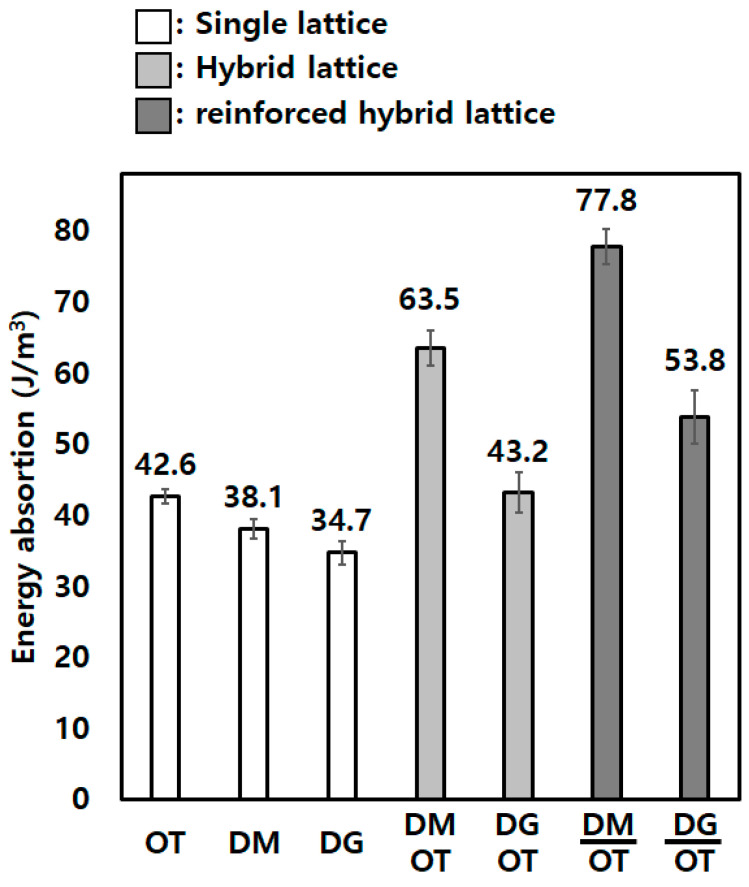
The value for energy absorption of single-, hybrid, and reinforced hybrid lattice structures.

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
