# Peer review of "Enhanced Energy Absorption of Additive-Manufactured Ti-6Al-4V Parts via Hybrid Lattice Structures"

_micromachines, 2023, doi:10.3390/mi14111982_

Round 1

Reviewer 2 Report

Thank you for submitting your manuscript titled " Enhanced energy absorption of additive manufactured Ti-6Al-4V parts via hybrid lattice structures" to Micromachines. The study presented in this manuscript demonstrates the enhanced energy absorption capability of hybrid lattice structures and emphasizes their potential in various industrial applications. By introducing hybrid lattice structures, the inherent energy absorption limitations of single lattice structures are effectively overcome. The comprehensive comparison between single lattice structures and hybrid lattice structures of equal density clearly showcases the superior energy absorption performance of the latter, highlighting the enormous potential of hybrid lattice structures and providing a promising approach to address energy absorption challenges in various industrial applications.

Overall, this manuscript is well-structured in the introduction section, the experimental design is reasonable, the discussion of the results provides sufficient interpretation of the experimental results, and the data analysis is reasonable. The figures are clear, representative, and effectively compare the results. However, from my perspective, there are also some issues that need to be addressed to improve the manuscript. I hope these comments will be helpful to the authors.

1. Introduction: The introduction section lacks a clear demonstration of the significance and importance of this study. It is recommended to provide a more detailed introduction to the problems in previous research and the limitations of these problems in the application of related research. This will better emphasize the importance of this study.

2. Experimental Design: The setting of the printing parameters based on the manufacturer's recommendations is not rigorous enough. Although the study focuses on the mechanical properties of different lattice structures, it is important to ensure that the printing is conducted under the same parameters. Deviations in the parameters may introduce some randomness in the formation of the parts and result in microstructural defects, which may affect the experimental results. It is suggested to optimize the process parameters before the experiment to reduce the influence of the formation defects unique to additive manufacturing.

3. Lattice Structure Design: The justification for placing the octagonal truss structure at the bottom of the structure is based on its structural stability within the unit cell. However, this justification is not rigorous enough with just one sentence. It would be better to provide specific experiments to demonstrate its stability or provide theoretical support.

4. Density Selection: The reason for choosing densities of 10%, 30%, and 50% is not explained. It would be helpful to provide a rationale for these specific density selections.

5. Results and Discussion: In the discussion of the compression behavior of the hybrid lattice structure, it is stated that a density of 30% is chosen because it exhibits the smallest deviations in compression strength and energy absorption. However, there is no specific data presented to support this justification. It is recommended to include specific data in the manuscript to support the chosen density.

6. Figure Presentation: In the results and discussion section, the figures for the hybrid lattice structures do not demonstrate the energy absorption values of the different structures. The figure presentation is also not consistent.

With these revisions, I recommend accepting this manuscript for publication in Micromachines.

While the manuscript is generally well-written, there are several instances throughout the text where grammatical errors, awkward phrasing, and imprecise language detract from the clarity and readability of the manuscript. I recommend a thorough proofreading and revision of the entire manuscript to address these issues.

I hope you find these comments helpful in improving the quality of the English language used in your manuscript. Once these revisions are made, I believe the manuscript will be suitable for publication in Micromachines.
